# The *Shigella* Vaccines Pipeline

**DOI:** 10.3390/vaccines10091376

**Published:** 2022-08-24

**Authors:** Calman Alexander MacLennan, Stephanie Grow, Lyou-fu Ma, Andrew Duncan Steele

**Affiliations:** Enteric & Diarrheal Diseases, Global Health, Bill & Melinda Gates Foundation, 500 5th Ave. N, Seattle, WA 98109, USA

**Keywords:** *Shigella*, shigellosis, flexneri, sonnei, vaccines, glycoconjugate, live attenuated, global health, diarrhea, dysentery

## Abstract

*Shigella* is the leading cause of global diarrheal deaths that currently lacks a licensed vaccine. Shigellosis drives antimicrobial resistance and leads to economic impact through linear growth faltering. Today, there is a robust pipeline of vaccines in clinical development which are broadly divided into parenteral glycoconjugate vaccines, consisting of O-antigen conjugated to carrier proteins, and oral live attenuated vaccines, which incorporate targeted genetic mutations seeking to optimize the balance between reactogenicity, immunogenicity and ultimately protection. Proof of efficacy has previously been shown with both approaches but for various reasons no vaccine has been licensed to date. In this report, we outline the requirements for a *Shigella* vaccine and describe the current pipeline in the context of the many candidates that have previously failed or been abandoned. The report refers to papers from individual vaccine developers in this special supplement of *Vaccines* which is focused on *Shigella* vaccines. Once readouts of safety and immunogenicity from current trials of lead candidate vaccines among the target population of young children in low- and middle-income countries are available, the likely time to licensure of a first *Shigella* vaccine will become clearer.

## 1. Introduction

Shigellosis is caused by an infection of the gastrointestinal tract with Gram-negative bacteria belonging to the genus *Shigella*. The pathogenesis of shigellosis involves the invasion of the colonic mucosa which results in inflammation. Clinical presentation is either with bloody diarrhea, otherwise known as dysentery, or watery diarrhea. Although dysentery is the classical presentation of shigellosis, watery diarrhea is the more common presentation. While dysentery is highly suggestive of shigellosis, presentation with watery diarrhea is clinically indistinguishable from the many other etiologies of watery diarrhea. Although human-restricted, shigellosis is highly transmissible by the fecal–oral route, contaminated food and water and fomites. It is estimated that 10 s to 100 s of bacteria are sufficient to cause disease [1]. *Shigella* is responsible for a large global burden of disease and is the leading bacterial cause of diarrheal deaths worldwide. Children under 5 years of age in low- and middle-income countries (LMICs) are most affected, particularly in the second year of life [2,3,4,5]. The Global Burden of Disease 2019 estimated 148,000 deaths from shigellosis, 93,000 of which were among children under five years of age [6]. Shigellosis is also a problem among travelers and military personnel deployed in endemic areas [7].

There are four species of *Shigella*, and three of these have multiple serotypes. This large number of *Shigella* serotypes presents a challenge for vaccine development, since immunologic protection is largely serotype-specific. *Shigella flexneri* is the most important species globally, particularly in low-income countries, and consists of 15 serotypes, the most common of which is *S. flexneri* 2a, followed by 3a and 6. *Shigella sonnei* has just one serotype and is the dominant species in countries that are undergoing industrial development. *Shigella dysenteriae* has 15 serotypes, the most significant being *Shigella dysenteriae* type 1, which was responsible for epidemic outbreaks with high-case fatality rates that seem to have disappeared in the 21st century. *Shigella boydii*, which has 19 serotypes, is responsible for a small minority of shigellosis cases and is mainly detected in South Asia [5].

In addition to the direct morbidity and mortality due to diarrhea, *Shigella* is a major cause of linear growth faltering among young children in LMICs [8,9]. Current WHO diarrhea treatment guidelines are for empirical antibiotics to only be given where presentation is with dysentery. Though dysentery has high specificity for shigellosis, the disease commonly presents with watery diarrhea, which cannot be clinically distinguished from other etiologies of watery diarrhea [10]. Otherwise, treatment is largely supportive with an oral rehydration solution (ORS). This means that the large majority children in LMICs with shigellosis do not receive antibiotics. Diagnosis requires stool culture which is unavailable in most LMIC healthcare settings and usually takes two to three days. There are no affordable point of care diagnostic tests for *Shigella*. Finally, even when initiated, the use of antibiotics for the treatment of *Shigellas* is becoming increasingly compromised by the rise in antimicrobial resistance among circulating *Shigella* isolates [11,12,13].

All of these factors make the development of a vaccine against *Shigella* a pressing global public health need. Several approaches have been adopted by vaccine developers for the prevention of shigellosis over the years. These broadly divide between oral whole cell approaches, consisting mostly of live attenuated but also inactivated vaccines, and parenteral subunit approaches, mostly glycoconjugates, which are designed to target the immune response to key *Shigella* antigens. There is evidence for efficacy with both of these vaccine approaches. In the 1960s, a live attenuated vaccine developed by David Mel of the Yugoslav Army was shown to be efficacious both among military recruits [14] and children aged 2–8 years [15] in former Yugoslavia. More recently, efficacy was demonstrated among Israeli military recruits of a *S. sonnei* O-antigen glycoconjugate vaccine developed by John Robbins and colleagues at the National Institutes for Health (NIH), USA [16]. In a subsequent study among Israeli children, protection was shown to extend down to three years of age [17].

Nevertheless, despite over a 100 years of *Shigella* vaccine development and a large numbers of clinical trials (Figure 1), there is no licensed vaccine against shigellosis. However, this situation is set to change in the coming decade with an improved understanding of the basis for protection against *Shigella* and several candidates set to advance to late-stage clinical trials. In this article, we will consider the requirements for a successful vaccine against *Shigella* and provide an overview of the candidates in the *Shigella* vaccines pipeline.

## 2. Requirements for an Effective *Shigella* Vaccine, Target Antigens and Immunity to *Shigella*

Key attributes for *Shigella* vaccines have recently been published by the WHO in a Preferred Product Characteristics (PPC) document [18]. A *Shigella* vaccine must be safe, immunogenic and efficacious against moderate-to-severe diarrhea (MSD) due to *Shigella* infection among the key target population of infants and young children under 5 years in LMICs. Efficacy should be achieved through a primary immunization schedule of one to two doses delivered during the first 12 months of life with or without an additional booster dose. The PPC document stipulates 60% or more efficacy against *Shigella* MSD caused by vaccine serotypes and a minimum duration of protection between 24 months and 5 years. A vaccine should be safe and not immunologically interfere when coadministered with other recommended vaccines, and should be stable for two years at 2–8 °C. It should be cost-effective so that price is not a barrier to access in LMICs. The PPC document includes both oral and injectable (intramuscular, intradermal and subcutaneous) routes of administration.

Non-human primate studies [19], controlled human infection model (CHIM) studies [20] and epidemiological studies of natural infection with *Shigella* [21] all provide strong evidence that protection against shigellosis is largely serotype specific. These findings implicate O-antigen of lipopolysaccharide (LPS) as being the critical antigenic target for vaccine development. Shigellosis typically confers approximately 70% protection against subsequent homotypic infection but not heterotypic infection. This has implications in relation to the valency of vaccine needed for broad protection. The serotyping of *Shigella* isolates from the Global Enterics Multicenter Study (GEMS) [2] suggests that a vaccine consisting of *Shigella flexneri* 2a, 3a and 6, as well as *Shigella sonnei* O-antigens could provide direct coverage against 64% of global *Shigella* isolates [22]. Coverage could increase to 88% due to cross-protection against *S. flexneri* serotypes which has been observed in guinea pigs [23].

Nonetheless, there is also a body of evidence implicating surface protein antigens common to multiple serotypes in protection against shigellosis. Most prominent among these are the Ipa proteins which form the needle and extracellular complexes of the *Shigella* type 3 secretion system (T3SS), particular IpaB, IpaC and IpaD [24]. The T3SS is essential for the uptake of *Shigella* into epithelial cells. A potential advantage of whole-cell approaches to vaccine development is the inclusion of Ipa proteins along with other potentially protective protein antigens.

The relevant immunological mediators and mechanisms of protection are likely multifactorial. The most recognized is serum IgG to LPS O-antigen which forms the main immunological response to parenterally administered glycoconjugate *Shigella* vaccines. Both phase 3 studies with NIH *S. sonnei* glycoconjugate vaccines found a strong association between serum O-antigen IgG and protective efficacy [16,17]. Subsequent analyses by Cohen provide good support for this marker being a correlate of protection against shigellosis [25]. In the context of oral live attenuated *Shigella* vaccines and natural infection with wild-type *Shigella*, fecal O-antigen IgA has been closely implicated in protection. This difference is likely indicative of parenteral vaccines inducing systemic immunity, in contrast to natural infection and oral vaccines directly inducing mucosal immunity within the gastrointestinal tract.

Recent data from *Shigella* CHIM studies suggest that serum IgG to IpaB could also be a correlate of protection [26]. Efforts are underway to develop a standardized ELISA for antibodies to O-antigens and establish a first international standard serum for these assays [27]. As well as gauging vaccine responses in terms of absolute levels of antibodies, there have been moves to determine the functionality of these antibodies focusing on the development of a serum bactericidal assay that can be transferred between laboratories [28,29]. Several studies have focused on cellular markers of immunity leading to antibody secreting cells (ASCs), B memory cells (B_M_), and CD4^+^ and CD8^+^ T cells being proposed as having roles in protection against shigellosis [30,31,32].

Several animal models have been used to study shigellosis and assess the potential protective efficacy of candidate vaccines. These include the mouse pulmonary infection [33], guinea pig conjunctivitis (Sereny test) [34] and guinea pig rectocolitis models [35]. However, since *Shigella* is a human-restricted pathogen and even non-human primates require much higher bacterial inocula to induce shigellosis compared to humans, all of these models have limitations. For this reason, there has been a long-standing emphasis on assessing candidate *Shigella* vaccines in humans. Aligned with this, the establishment of the *Shigella* CHIM at three centers in the United States, has permitted the early assessment of vaccine efficacy in humans [36,37,38]. Consensus has been reached among these centers in relation to the overall conduct [39], clinical endpoints [40] and laboratory investigation priorities [41] for these studies.

## 3. Oral Whole Cell Vaccines

Oral whole cell vaccines can be divided between whole cell killed (Table 1) and live attenuated approaches (Table 2).

### 3.1. Whole Cell Killed Vaccines

The earliest vaccines, beginning with the first attempt at a *Shigella* vaccine by Kiyoshi Shiga, adopted the whole-cell killed approach [42]. Key to their development was ensuring that inactivation did not adversely affect the structure of relevant antigens. Ether, formalin and heat were all utilized as means of inactivation. In general, the early whole cell killed vaccines resulted in high levels of reactogenicity. They were tested in poorly designed studies that prevented any meaningful conclusion about efficacy [42].

Relatively recently, formalin-inactivated *S. sonnei* (SsWc) [43] and *S. flexneri* 2a (Sf2aWC) [44] monovalent whole cell vaccine candidates were developed by the Walter Reed Army Institute for Research (WRAIR) and tested in a phase 1 study in the US. The vaccines were well tolerated though immune responses were variable. While a trivalent version of the vaccine was subsequently developed [45], it has not been tested in a clinical trial. The whole cell killed approach has now largely been abandoned.

**Table 1 vaccines-10-01376-t001:** ***Shigella*** **whole** **cell** **killed** **vaccines**.

Name	Description	Developer	Species/Serotype and Strain	Genetic Deletions	Status	Furthest Stage of Development	Reference
SsWc	*S. sonnei* monovalent formalin-inactivated	WRAIR	*S. sonnei* Moseley	None	Discontinued	Phase 1	[43]
Sf2aWC	*S. flexneri* 2a monovalent formalin-inactivated	WRAIR	*S. flexneri* 2a 2457T	None	Discontinued	Phase 1	[44]

### 3.2. Live Attenuated Vaccines

#### 3.2.1. Mel and Streptomycin-Dependent Vaccines

Studies by Mel and colleagues from the 1960s and 1970s [14,15] paved the way for the development of a large number of live attenuated vaccines over the following fifty years. The original Mel vaccines were developed by passaging *Shigella* strains using media containing the antibiotic streptomycin. This resulted in the strains becoming both streptomycin dependent (SmD) and attenuated, though the exact genetic mutations leading to attenuation were unknown. Although SmD *Shigella* vaccines were tested in subjects over multiple studies in Yugoslavia and the US, they had a number of issues. First, the primary immunization schedule consisted of four doses. Second, protection was relatively short-lived, lasting for a year. Third, some vaccine lots lost their streptomycin dependence. Finally, there were issues with manufacturing. These vaccines were never licensed and were eventually abandoned.

#### 3.2.2. Vadizen and T_32_ Vaccines

Around the same time, another live attenuated vaccine, the Vadizen live vaccine, was developed in Romania by Istrati from the *S. flexneri* 2a T_32_ strain by multiple passages on nutrient agar [46]. During a period of five years, this was tested in over 36,000 subjects in 12 countries. Protection of 86.6% was reported against dysentery in children with protection lasting for 6 months [47]. Of note, the Lanzhou Institute of Biological Products developed the F2a-sonnei (FS) bivalent vaccine based on the *S. flexneri* 2a T_32_ strain containing an *S. sonnei* plasmid. This enabled the co-expression of O-antigens of *S. flexneri* 2a and *S. sonnei*. In field studies in China, FS was reported to give 61–65% protection against *S. flexneri* 2a and 57–72% *S. sonnei* shigellosis [48]. The current development status of this vaccine construct is unknown.

More recently, the advent of whole genome sequencing enabled the development of well-defined live attenuated *Shigella* vaccines with targeted genetic mutations. However, similar challenges are faced by all live attenuated vaccine candidates in balancing acceptable levels of reactogenicity with sufficient immunogenicity to confer protection. Below, we report on a selection of such vaccines that have been tested in clinical trials.

#### 3.2.3. Center for Vaccine Development and guaBA-Deficient Vaccines

The Center for Vaccine Development at the University of Maryland has developed a number of live attenuated *Shigella* vaccines based on the well-characterized *S. flexneri* 2a 2457T strain. CVD 1203 contained mutations in the *aroA* and *virG* genes that encode proteins involved in amino acid biosynthesis and intra- and intercellular motility. Although immunogenic, the vaccine was excessively reactogenic, causing dysentery in 72% of recipients in a phase 1 study [49] and was discontinued. For CVD 1207, *guaBA*, *virG*, *set* and *sen* genes were deleted. *guaBA* is required for guanine biosynthesis, while *set* and *sen* encode the ShET1 and ShET2 *Shigella* enterotoxins, respectively. This new approach resulted in tolerability in a phase 1 study but limited immunogenicity [50].

Further iterations of CVD 1207 maintained the *guaBA* mutation but reverted to using wild-type *virG*, either with intact (CVD 1204) or deleted (CVD 1208) *sen* and *set* genes. The deletion of the *Shigella* enterotoxins was shown to be required since CVD1204 resulted in unacceptable reactogenicity [51]. CVD 1208S (CVD 1208 grown on animal-free media) advanced to a phase 2 study with the intent to challenge with wild-type 2457T. However, the trial was terminated following the recruitment of 20 subjects due to reactogenicity [52].

#### 3.2.4. Pasteur Institute and *virG*-Deficient Vaccines

Another challenge facing development of live attenuated *Shigella* vaccines is the difference in reactogenicity and immunogenicity of the same vaccine among different populations. SC602 is a *S. flexneri* 2a candidate vaccine developed at the Pasteur Institute, Paris. It is based on the wild-type *S. flexneri* 2a 494 strain with deletions in *virG* and *iuc* which encodes the siderophore, aerobactin. At 10^4^ cfu, SC602 was mildly reactogenic in US adults but immunogenic and protected against fever, dysentery and severe symptoms following challenge with *S. flexneri* 2a 2457T [53]. When tested in Bangladeshi adults and children, the vaccine was minimally reactogenic but induced only a limited immune response [54].

The poor immunogenicity of live attenuated vaccines has also been observed among LMIC populations with licensed vaccines against cholera, rotavirus and polio. Multiple reasons may account for this, a key one being environmental enteric dysfunction (EDD) which commonly occurs in children in LMICs [55]. EDD is an acquired enteropathy of the small intestine. It is thought to be caused by subclinical infections and is characterized by inflammation and blunting of the villi. EDD is associated with systemic inflammation and malnutrition. Reducing the time taken to test candidate vaccines in LMIC children is valuable for an early understanding of which candidate vaccines should progress.

The Pasteur Institute also developed SC599, a live attenuated *Shigella* vaccine candidate based on wild-type *Shigella dysenteriae* type 1 strain SC595 with deleted *virG*, *ent* and *fep* genes, the latter two genes encoding iron chelation proteins. Although well-tolerated in phase 1 trials, the vaccine was poorly immunogenic and has not progressed [56].

#### 3.2.5. National Bacteriology Laboratory, Stockholm, and aroD-Deficient Vaccines

In the early 1990s, the National Bacteriology Laboratory in Stockholm developed live *Shigella* vaccine candidates attenuated through the deletion of *aroD* which made them auxotrophic for aromatic compounds. SFL124 was based on the moderately virulent parent strain *S. flexneri* SFL1. The candidate was well tolerated in phase 1 studies in adults in Sweden [57] and Vietnam [58], and in a phase 2 studies in Vietnamese children [59], but showed variable immunogenicity.

Subsequently, SLF1070 was developed with *aroD* deleted from the more virulent *S. flexneri* 2a 2457T strain. When tested in a phase 1 study in Swedish adults, a narrow safety-immunogenicity profile was demonstrated [60]. Neither candidate progressed into later clinical trials.

#### 3.2.6. WRAIR and the WRSS Vaccines

WRAIR developed a series of live attenuated *S. sonnei* vaccines based on the wild-type Moseley *S. sonnei* strain with *virG* deleted. A common problem with *S. sonnei* strains is loss of the virulence plasmid. However, the plasmid is relatively stable in the Moseley strain. WRSS1, the first-generation vaccine of this series, was immunogenic but resulted in diarrhea and fever among US and Israeli adult volunteers in phase 1 and phase 2 studies [61,62]. WRSS1 was better tolerated when tested in a phase 1 study in Bangladeshi adults and children, but immune responses were modest, requiring multiple doses, and were of short duration [63].

Further attenuating mutations were introduced to address the reactogenicity resulting in the WRSS2 and WRSS3 candidate vaccines. While both have deleted enterotoxin genes *senA* and *senB*, WRSS3 has *msbB* deleted from the virulence plasmid. The removal of *msbB*, which encodes an acyltransferase, results in the loss of an acyl chain from the lipid A of LPS with a consequent reduction in reactogenicity [64]. Both vaccines were immunogenic and reasonably tolerated in a phase 1 trial in the US [65].

#### 3.2.7. Typhoid *Shigella* Combination Vaccine

An innovative approach to *Shigella* vaccine development was the use of the licensed typhoid Ty21a live attenuated vaccine as a vector for the delivery of *Shigella sonnei* O-antigen. The 5076-1C vaccine was developed by Formal at the University of Maryland in the 1980s [66]. Though well tolerated [67], it suffered from inconsistency in production leading to variability in immunogenicity and protection afforded in CHIM studies [36,68]. Further development of the vaccine was abandoned.

#### 3.2.8. *E. coli*—*Shigella* Combination Vaccines and ShigETEC

In order to improve the tolerability of live attenuated *Shigella* vaccines, in a similar approach to the Ty21a *Shigella* candidate 5076-1C, key *Shigella* antigens have been expressed in *E. coli*. PGAI 42-1-15 was based on *E. coli* O8 genetically modified to express *S. flexneri* 2a O-antigen [69]. Though well tolerated, the vaccine failed to protect in a *S. flexneri* 2a CHIM study [70]. Similar vaccines, EcSf2a-1 and ECSf2a-2, were developed using *E. coli* K12 that could express *S. flexneri* 2a O-antigen and also invade epithelial cells due to the inclusion of the invasion plasmid from *S. flexneri* 5, with *aroD* deleted from ECSf2a-2 to reduce reactogenicity. Unacceptable reactogenicity was a problem for ECSf2a-1 [71], while ECSf2a-2 failed to protect in a CHIM study [72]. This line of approach was subsequently discontinued.

An intriguing and almost opposite approach to *E. coli* engineered to express *Shigella* antigens, is the ShigETEC live attenuated vaccine being developed by Eveliqure, Vienna [73]. ShigETEC is based on wild-type *S. flexneri* 2a with *Shigella* O-antigen and Ipa antigens, the best-characterized targets of protective immunity, removed through the deletion of *rfbF* and *ipaBC*, respectively, together with the deletion of *setAB* to remove enterotoxins. The vaccine was engineered to express a fusion protein of LT and ST, the labile and stable toxins of enterotoxigenic *E. coli* (ETEC), hence the name ShigETEC. The vaccine was well tolerated and immunogenic in a recent phase 1 trial in Hungary and is set to progress into phase 2 studies [73]. A key open question is which antigens will induce protective immunity against *Shigella*.

Similar to the ShigaETEC approach, a new ambitious *Shigella* vaccine strategy at the Center for Vaccine Development is to develop a hexavalent *Shigella* ETEC live attenuated vaccine consisting of six *Shigella* strains (*S. flexneri* 1b, 2a, 3a and 6, *S. sonnei* and *S. dysenteriae* 1) engineered to express ETEC antigens. This is currently in preclinical development [74].

**Table 2 vaccines-10-01376-t002:** ***Shigella*** **live** **attenuated** **vaccines**.

Name	Description	Developer	Species/Serotype and Strain	Genetic Deletions	Status	Furthest Stage of Development	References
SmD	Streptomycin-dependent *Shigella* strains	Yugoslav Army	Various	Not known	Discontinued	Phase 3	[14,15]
Vadizen live vaccine	*S. flexneri* 2a passaged on nutrient agar	Istrati	*S. flexneri* 2a T_32_ Istrati	Not known	Discontinued	Phase 3	[46,47]
F2a-sonnei	*S. flexneri* 2a LAV expressing *S. sonnei* O-antigen	Lanzhou Institute	*S. flexneri* 2a T_32_	Not known	Unknown	Phase 3	[48]
CVD1203	*S. flexneri* 2a monovalent LAV	University of Maryland	*S. flexneri* 2a 2457T	Δ *aroA* and *virG*	Discontinued	Phase 1	[49]
CVD1204	*S. flexneri* 2a monovalent LAV	University of Maryland	*S. flexneri* 2a 2457T	Δ *guaBA*	Discontinued	Phase 1	[51]
CVD1207	*S. flexneri* 2a monovalent LAV	University of Maryland	*S. flexneri* 2a 2457T	Δ *guaBA*, *virG*, *set* and *sen*	Discontinued	Phase 1	[50]
CVD1208S	*S. flexneri* 2a monovalent LAV grown on animal-free media	University of Maryland	*S. flexneri *2a 2457T	Δ *guaBA*, *set* and *sen*	Discontinued	Phase 2	[52]
SC602	*S. flexneri* 2a monovalent LAV	Pasteur Institute	*S. flexneri* 2a 494	Δ *virG* and *iuc*	Discontinued	Phase 2	[53,54]
SC599	*S. dysenteriae* type 1 monovalent LAV	Pasteur Institute	*S. dysenteriae type* 1 SC595	Δ *virG*, *ent*, and *fep*	Discontinued	Phase 1	[56]
SFL124	*S. flexneri *2a monovalent LAV	National Bac. Lab, Stockholm	*S. flexneri* 2a SFL1	Δ *aroD*	Discontinued	Phase 2	[57,58,59]
SFL1070	*S. flexneri* 2a monovalent LAV	National Bac. Lab, Stockholm	*S. flexneri* 2a 2457T	Δ *aroD*	Discontinued	Phase 1	[60]
WRSS1	*S. sonnei* monovalent LAV	WRAIR	*S. Sonnei* Moseley	Δ *virG*	Discontinued	Phase 2	[61,62,63]
WRSS2	*S. sonnei* monovalent LAV	WRAIR	*S. Sonnei* Moseley	Δ *virG*, *senA*, and *senB*	Active	Phase 1	[64,65]
WRSS3	*S. sonnei* monovalent LAV	WRAIR	*S. Sonnei* Moseley	Δ *virG*, *senA*, *senB*, and *msbB*	Active	Phase 1	[64,65]
5076-1C	*Salmonella* Typhi LAV expressing *S. sonnei* O-antigen	University of Maryland	*Salmonella* Typhi Ty21a	Δ *galE*	Discontinued	Phase 2	[36,66,67,68]
PGAI 42-1-15	*E. coli* LAV expressing *S. flexneri* 2a O-antigen	University of Maryland	*E. coli* O8 RJ 91	Not known	Discontinued	Phase 2	[69,70]
EcSf2a-1	*E. coli* LAV with *S. flexneri* 5 invasion plasmid, expressing *S. flexneri* 2a O-antigen	University of Maryland	*E. coli* K-12 395-1		Discontinued	Phase 1	[71]
EcSF2a-2	*E. coli* LAV with *S. flexneri* 5 invasion plasmid, expressing *S. flexneri* 2a O-antigen	University of Maryland	*E. coli* K-12 395-1	Δ *aroD*	Discontinued	Phase 2	[71,72]
ShigETEC	*S. flexneri *2a LAV expressing fusion protein of B subunit of ETEC heat-labile and heat-stable enterotoxins	Eveliqure	*S. flexneri* 2a 2457T	Δ *rfbF*, *ipaBC*, and *setBA*	Active	Phase 1	[73]

## 4. Subunit Vaccines

### 4.1. Glycoconjugate Vaccines

#### 4.1.1. Cohen and the NIH Glyconjugate Vaccines

The glycoconjugate approach to *Shigella* vaccines builds on the successful use of this technology for vaccines against the encapsulated bacteria *Haemophilus influenzae* b, meningococcus, pneumococcus and, more recently, *Salmonella* typhi. With the exception of *S. sonnei* (which has a capsule of O-antigen [75]), *Shigella* bacteria lack capsules, and so the glycan exploited in these vaccines is the O-antigen of LPS. As mentioned previously, serotype-specific protection following shigellosis implicates the immune response to O-antigen (which is serotype-specific) in protection. This is the key epidemiological observation on which the *Shigella* glycoconjugate vaccines are based.

A large body of work saw the relatively rapid advancement of glycoconjugates developed by Robbins and colleagues at the NIH through to efficacy studies in Israel. These prototype glycoconjugate vaccines consisted of O-antigen from *S. sonnei* or *S. flexneri* 2a chemically linked to recombinant exoprotein A of *Pseudomonas aeruginosa* (rEPA) (Table 3). The vaccines were clinically evaluated in Israel by Cohen and colleagues who had been studying immunity to *Shigella* and had collected strong observational evidence for the importance of serum O-antigen IgG in protection against shigellosis [76]. The safety and immunogenicity of *S. sonnei* O-antigen/rEPA were demonstrated in phase 1 and phase 2 clinical trials leading to a pivotal phase 3 study which consisted of three arms with volunteers receiving *S. sonnei* O-antigen/rEPA, the EcSf2a-2 live attenuated *S. flexneri* vaccine, or placebo [16].

The finding of 74% efficacy against shigellosis due to *S. sonnei* in the arm receiving *S. sonnei* O-antigen/rEPA was remarkable considering this was after a single dose of vaccine with participants followed up for a duration of two years. The lack of cases of shigellosis due to *S. flexneri* 2a in the study prevented any conclusions being made on the efficacy of EcSf2a-2. Given that study findings were published 25 years ago, it is surprising that no glycoconjugate vaccine has yet been licensed. There are perhaps several reasons for this.

While protection was demonstrated in military recruits in Israel, the key target population for a global *Shigella* vaccine is young children in LMICs. Thirteen years later, in 2010, findings from a second efficacy study in Israel were published by Paswell and colleagues [17], this time with children receiving either *S. sonnei* O-antigen/rEPA or *S. flexneri* 2a O-antigen/rEPA. While protection was demonstrated against *S. sonnei* shigellosis among children aged 3–4 years, protection was not seen in children under three years of age. Loss of protection corresponded to a decrease in serum O-antigen IgG titer further supporting a role for this modality of immunity in protection against shigellosis. Again, insufficient cases of *S. flexneri* 2a shigellosis precluded any conclusions about the efficacy of *S. flexneri* 2a O-antigen/rEPA. Neither *S. sonnei* O-antigen/rEPA nor *S. flexneri* 2a O-antigen/rEPA were ever commercialized.

#### 4.1.2. LimmaTech and Bioconjugate Vaccines

Limmatech Biologics AG, based in Zurich, developed an innovative glycoconjugate technology known as bioconjugation where *E. coli* bacteria are genetically engineered to synthesize and chemically couple glycans from exogenous bacteria to protein carriers, exploiting the oligosaccharide transferase PglB [77]. As proof of principle, an *S. dysenteriae* type 1 bioconjugate, GVXN SD133-EPA was produced with O-antigen from *S. dysenteriae* type 1 linked to genetically detoxified exotoxin protein A of *Pseudomonas aeruginosa* and was found to be safe and immunogenic in a phase 1 study [78].

Subsequently, the technology was used to develop Flexyn2a-EPA, a vaccine against *S. flexneri* 2a which, following promising immunogenicity in a phase 1 study in the US, was assessed for efficacy against shigellosis in a CHIM study at Johns Hopkins University. Although the vaccine failed to meet the primary endpoint of protection against all forms of shigellosis, it did protect against more severe shigellosis in a post hoc analysis [38]. This prompted a series of international expert workshops to harmonize the conduct, clinical endpoints and immunological assays of future CHIMs to facilitate comparison between vaccines in development. The resulting consensus documents were published in *Clinical Infectious Diseases* (2019 supplement 8). As with the NIH *S. sonnei* vaccine, protection was closely associated with serum IgG to *S. flexneri* O-antigen [38,79]. On the basis of good immunogenicity in the phase 1 study and protection against more severe shigellosis in the CHIM study, a four-valent vaccine, S4V-EPA, was developed consisting of bioconjugates against *S. sonnei*, *flexneri* 3a and *flexneri* 6, as well as *S. flexneri* 2a. This is currently completing an age-descending dose-finding study in Kenya [77].

#### 4.1.3. Pasteur Institute and Synthetic O-Antigen Conjugate Vaccines

Separate work by Pozgay and Robbins [80,81] with different *Shigella* conjugate vaccine constructs demonstrated that the use of O-antigens of reduced length could result in an enhanced serum IgG response to O-antigen compared with conjugates employing native length O-antigen. This observation provided supporting justification for the synthetic approach to generate *Shigella* O-antigens for conjugation at the Pasteur Institute. Using an elaborate process involving multiple chemical steps, a series of *S. flexneri* 2a synthetic O-antigens were generated and conjugated to tetanus toxoid prior to testing in mice for immunogenicity.

Evaluation in animals led to the selection of a candidate vaccine, SF2a-TT15, with O-antigen consisting of three pentasaccharide repeating units (giving a total of 15 saccharides). This vaccine was tested by Cohen in Israeli adults and found to be highly immunogenic, even after a single dose [82]. As with the Limmatech quadrivalent vaccine, SF2a-TT15 is currently being evaluated in an age-descending dose-finding study in Kenya and an adult CHIM study at the Center for Vaccine Development in Baltimore [83].

#### 4.1.4. Beijing Zhifei Lvzhu Biopharmaceuticals Bivalent Glycoconjugate Vaccine

Using more conventional glycoconjugate technology, Beijing Zhifei Lvzhu Biopharmaceuticals designed and developed a bivalent vaccine, ZF0901, using O-antigen from *S. sonnei* and *S. flexneri* 2a conjugated to tetanus toxoid with adipic acid dihydrazide as linker. Following a phase 1 descending age study in China to show safety [84], ZF0901 was tested in a phase 2 age-descending study in which it was found to be safe and immunogenic, and is currently being tested in a phase 3 study [85].

### 4.2. Other Subunit Vaccines

#### 4.2.1. WRAIR, Invasin Complex and Proteosome Complex Vaccines

An interesting alternative subunit vaccine approach to the glycoconjugates is the invasin complex or Invaplex technology developed by WRAIR [86]. Essentially, rather than conjugating O-antigen to carrier protein, Invaplex consists of a physical mixture of *Shigella* LPS and Ipa proteins. The initial iteration of Invaplex, native Invaplex or Invaplex_NAT_, consisted of a complex of Ipa antigens (IpaB, IpaC and IpaD) and LPS extracted from wild-type *S. flexneri* 2a. Three doses delivered intranasally were well tolerated and immunogenic in phase 1 studies [87,88] in the US but failed to protect in a CHIM study [89].

Subsequently, a more defined version of Invaplex, artificial Invaplex or Invaplex_AR_, was produced using recombinant IpaB and IpaC generated in *E. coli*, and LPS from *S. flexneri* 2a which was safe but not consistently immunogenic when delivered intranasally to volunteers in a phase 1 study [86]. As a further iteration, artificial detoxified Invaplex (Invaplex_AR-Detox_) employs an LPS of low reactogenicity purified from *S. flexneri* 2a with deleted *msbB* genes. This has allowed the parenteral administration of Invaplex_AR-Detox_ with good safety and immunogenicity results in a recent phase 1 study [86]. It is surprising that high antibody titers are induced by the O-antigen of LPS in the absence of conjugation. The added protective value and extent of cross-protection resulting from the inclusion of Ipa proteins in the vaccine is currently not clear, although existing data from animals and humans suggest that the immune response to IpaB and IpaC will enhance protection.

In a similar but earlier approach to Invaplex, WRAIR complexed LPS from *S. flexneri* 2a with proteosomes (outer membrane proteins) from meningococcus to give a Proteosome-*Shigella flexneri* 2a lipopolysaccharide vaccine. This vaccine was only tested in phase 1 and was administered intranasally [90]. It was well tolerated and induced a modest immune response to O-antigen.

#### 4.2.2. GSK Vaccines Institute for Global Health and Outer Membrane Vesicle Vaccines

A different subunit approach which has similarities with whole cell killed vaccines is the use of bacterial native outer membrane vesicles (OMVs) as vaccines. The GSK Vaccines Institute for Global Health (GVGH) adopted this approach by increasing the spontaneous release of OMV (referred to as ‘GMMA’—Generalised Modules for Membrane Antigens, by GVGH), which are blebs of outer membrane from the surface of *Shigella*, by the deletion of *tolR* [91]. *tolR* is a gene involved in maintaining the integrity of the connection between the inner and outer membranes in Gram-negative bacteria. In addition, *htrB*, which, like *msbB*, encodes an acyl transferase, was deleted to reduce reactogenicity, and *virG* to prevent epithelial invasion. This technology has the advantage of being straightforward to deploy with the potential to produce vaccine at low cost. GVGH considers the vesicles to be a vehicle for the delivery of O-antigen. Another advantage of this approach is that multiple other outer membrane components are presented to the immune system, including many outer membrane proteins [92], though not Ipa proteins, likely because the T3SS is tethered to the bacterial inner membrane.

A first OMV candidate vaccine, 1790GAHB, was generated from *S. sonnei* 53G with the above mutations. Though well tolerated, it was poorly immunogenic compared with historic *Shigella* glycoconjugate vaccines whether administered intramuscularly, intradermally or intranasally in European adults [93]. This is thought to be due to low content of LPS O-antigen with only 10% of LPS molecules in 1790GAHB expressing O-antigen. Although the vaccine was able to increase pre-existing high titers of serum O-antigen IgG in Kenyan adults [94], it failed to protect in a CHIM study in Cincinnati [95]. A quadrivalent vaccine, altSonflex1-2-3, consisting of OMV from *S. flexneri* 1b, 2a and 3a and *S. sonnei* (with higher O-antigen expression than 1790GAHB) is currently being tested in a phase 1/2 clinical trial [92].

**Table 3 vaccines-10-01376-t003:** ***Shigella*** **subunit** **vaccines**.

Name	Description	Developer	Species/Serotype and Strain	Genetic Deletions	Status	Furthest Stage of Development	References
*S. sonnei* O-antigen/rEPA	*S. sonnei* monovalent O-antigen glycoconjugate	NIH	*S. sonnei*	-	Discontinued	Phase 3	[16,76]
*S. flexneri* 2a O-antigen/rEPA	*S. flexneri* 2a monovalentO-antigen glycoconjugate	NIH	*S. flexneri* 2a	-	Discontinued	Phase 3	[17,76]
S4V-EPA	Quadrivalent O-antigen bioconjugate	LimmaTech	*S. flexneri* 2a, 3a, 6, and *S. sonnei*	-	Active	Phase 2	[38,77,79]
GVXN SD133-EPA	*S. dysenteriae* type 1 monovalent O-antigen bioconjugate	LimmaTech	*S. dysenteriae* type 1	-	Discontinued	Phase 1	[77,78]
SF2a-TT15	*S. flexneri* 2a synthetic O-antigen conjugate	Pasteur Institute	*S. flexneri* 2a	-	Active	Phase 2	[82,83]
ZF0901	Bivalent O-antigen glycoconjugate	Beijing Zhifei Lvzhu Biopharmaceuticals	*S. flexneri* 2a and *S. sonnei*	-	Active	Phase 3	[84,85]
Invaplex_NAT_	Natural *S. flexneri* 2a invasin complex (LPS, IpaB, IpaC, and IpaD)	WRAIR	*S. flexneri* 2a 2457T	-	Discontinued	Phase 2	[86,87,88,89]
Invaplex_AR_	Artificial *S. flexneri* 2a invasin complex (LPS, IpaB and IpaC)	WRAIR	*S. flexneri* 2a 2457T	-	Discontinued	Phase 1	[86]
Invaplex_AR-DETOX_	Artificial detoxified *S. flexneri* 2a invasin complex (recombinant IpaB and IpaC)	WRAIR	*S. flexneri* 2a 2457T	LPS from *S. flexneri* 2a Δ *msbB*	Active	Phase 1	[86]
Proteosome *Shigella flexneri* 2a LPS	*S. flexneri* 2a LPS and meningococcal outer membrane proteins	WRAIR	*S. flexneri* 2a	-	Discontinued	Phase 1	[90]
1790GAHB	Monovalent *S. sonnei* native outer membrane vesicle	GVGH (GSK)	*S. sonnei*	Δ *tolR*, *htrB*, and *virG*	Discontinued	Phase 2	[91,92,93,94,95]
altSonflex1-2-3	Quadrivalent *Shigella* native outer membrane vesicle	GVGH (GSK)	*S. flexneri* 1b, 2a, 3a, and *S. sonnei*	Δ *tolR*, *htrB*, and *virG*	Active	Phase 2	[91]

## 5. Discussion

Despite 100 years of vaccine development, there is to date no licensed vaccine to protect against shigellosis, the main cause of childhood bacterial diarrheal death globally. This is despite numerous candidate vaccines having been assessed in clinical trials and clinical efficacy proven with both the early live attenuated approaches of Mel and the Yugoslav Army in the 1960s, and the glycoconjugate approach of Robbins and the NIH in the 1990s. As indicated in this report, the large majority of vaccines tested in humans to date have been live attenuated vaccines. Their lack of success consistently comes down to the difficulty of balancing reactogenicity with sufficient immunogenicity to induce protection.

Where protection was demonstrated with the Mel SmD vaccines, this required multiple doses (a primary series of four doses) and duration of protection was limited to one year. However, according to the recently published WHO Preferred Product Characteristic for *Shigella* vaccines [18], a *Shigella* vaccine should induce protective immunity for at least two years after no more than two doses. The likely need for multivalent *Shigella* vaccines to provide sufficient breath of coverage against global circulating *Shigella* strains increases the challenge faced by live attenuated candidates. To date, it has proven exceedingly difficult to achieve an acceptable balance of reactogenicity and immunogenicity with monovalent *Shigella* vaccines.

Glycoconjugate technology appears more promising than live attenuated approaches for a licensed *Shigella* vaccine. Proof of efficacy was demonstrated by Cohen with the NIH *S. sonnei* O-antigen-rEPA candidate 25 years ago in young Israeli adults [16] and subsequently in children down to three years of age [17]. It is both surprising and disappointing that so few *Shigella* conjugate vaccines have entered clinical development since 1997. This is perhaps partly a result of the emphasis on pursuing the live attenuated approach by many vaccine developers during the 1990s and 2000s at a time when limited funding was available for *Shigella* vaccine development.

Following the disappointing results with so many live attenuated candidates, it is reassuring to see a balance across the global portfolio of *Shigella* vaccines with advancing clinical development of several promising glycoconjugate vaccines, as well as live attenuated vaccines advancing. The advent of molecular determination in recent years of pathogen etiology in diarrhea cases in place of exclusive reliance on stool culture [3] has increased the global awareness of the huge public health burden caused by *Shigella*. Growing worldwide concern about the threat of antimicrobial resistant *Shigella* [96] and the value of vaccines in curbing AMR [97], as well as an appreciation that *Shigella* is a major cause of linear growth faltering [98], are providing added impetus for *Shigella* vaccine development. These factors, together with increased funding for *Shigella* vaccines and the strong recommendation by the WHO’s Product Development Vaccine Advisory Committee (PDVAC) of the need to develop *Shigella* vaccines [99], are serving to accelerate the development of promising candidates.

Although glycoconjugate candidates rarely cause issues with reactogenicity compared with live attenuated vaccines, there remains the challenge of whether current candidates will prove to be sufficiently immunogenic among young children in LMICs. The most advanced glycoconjugate vaccine is the quadrivalent bioconjugate candidate, S4V-EPA, from Limmatech. This vaccine is currently completing its phase 2 study in young children in Kenya with the expectation of a readout of interim immunogenicity this year [77]. Meanwhile, the Pasteur Institute *S. flexneri* 2a synthetic O-antigen conjugate vaccine, SF2a-TT15, is completing an age descending, dose-finding study in Kenya [83]. The GVGH quadrivalent OMV vaccine, altSonflex-1-2-3, is in clinical evaluation in European adults [92].

The cumulative clinical data from these trials will provide much clarity as to whether a licensed *Shigella* vaccine is a realistic prospect in the next few years. Should these vaccines turn out to be safe but insufficiently immunogenic, there is the possibility of enhancing immune responses using adjuvants. To date, little is known about which adjuvants may potentiate current candidate *Shigella* vaccines, particularly the glycoconjugates. The use of adjuvants in the clinical trials of these vaccines has so far been limited to alum and effects have been inconsistent. This represents a knowledge gap in *Shigella* vaccine development. The application of our understanding of new adjuvants from the COVID-19 pandemic may help address this gap but requires formulation, stability and immunogenicity studies with lead candidate vaccines to begin now.

Although there is strong evidence that serum IgG to O-antigen is a correlate of protection in adults [25], this is yet to be shown in young children in LMICs. Consequently phase 3 efficacy studies are likely to be needed for licensure and WHO prequalification of *Shigella* vaccines for the global pediatric market. Phase 3 studies will be lengthy and costly but ultimately, if able to demonstrate efficacy and confirm the correlate of protection status for O-antigen IgG in children, may permit the licensure of subsequent vaccines on the basis of safety and non-inferior immunogenicity. As a critical enabling activity of this, work is underway at NIBSC to develop a first *Shigella* International Standard Serum and harmonize *Shigella* ELISAs [27].

Studies are currently underway to facilitate and expedite the initiation of phase 3 trials for the most promising candidate vaccines when ready [100]. Each of the candidate vaccines described ultimately requires a manufacturer to bring them to market. The lack of commercial incentive of a *Shigella* vaccine for the global pediatric market is unlikely to entice one of the big five multinational vaccine companies to manufacture a *Shigella* vaccine. It is more likely that companies from the Developing Country Vaccine Manufacturers Network [101] will partner to produce and, if successful at phase 3, license the most promising *Shigella* vaccines.

## 6. Conclusions

Over the past 100 years, *Shigella* vaccine development has become a graveyard filled with numerous candidates that entered clinical trials and proved to be either too reactogenic, insufficiently immunogenic or both. Nevertheless, proof of efficacy has been attained on more than one occasion and there are currently multiple promising candidates in clinical development, with lead candidates due to read out shortly from studies in the target populations of young children in LMICs.

## Figures and Tables

**Figure 1 vaccines-10-01376-f001:**
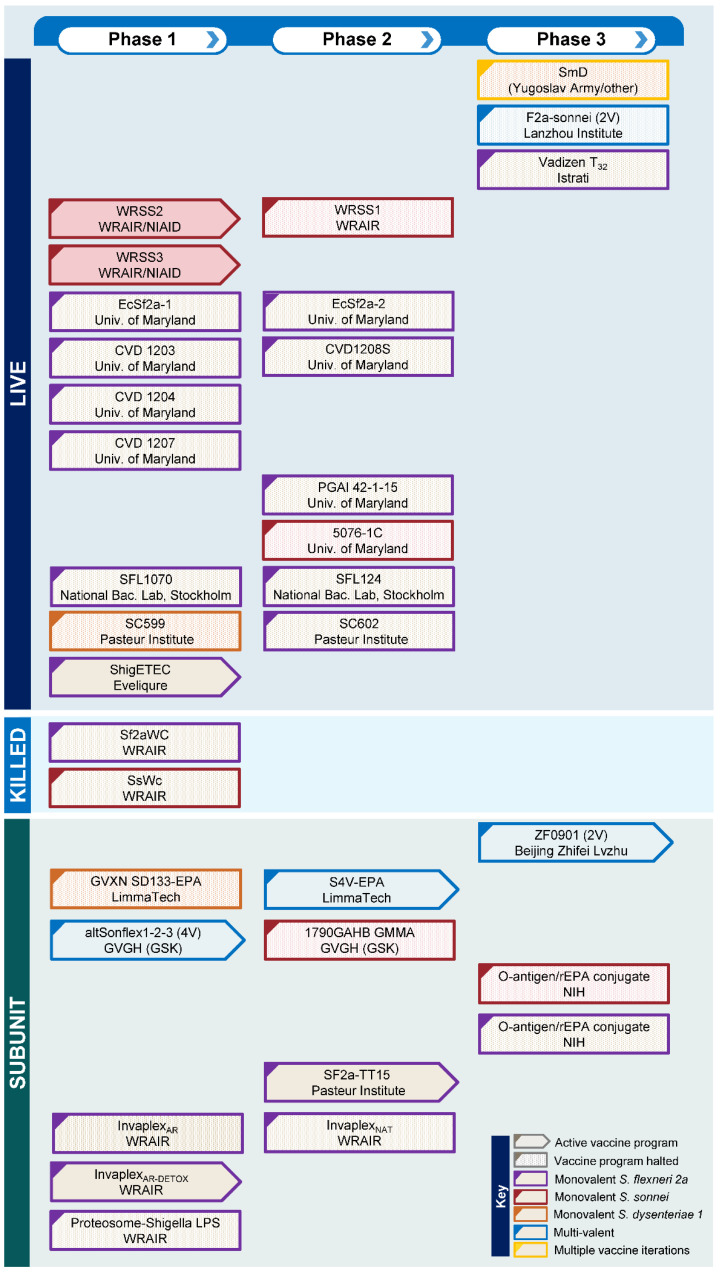
***Shigella* vaccines pipeline** indicating active and halted vaccine programs from the past 60 years by vaccine type and phase of clinical development.

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
