# Peer review of "The Shigella Vaccines Pipeline"

_vaccines, 2022, doi:10.3390/vaccines10091376_

Round 1

Reviewer 1 Report

The review entitled “The Shigella Vaccines Pipeline” by Calman A. MacLennan et al provides a general overview on past and current Shigella vaccines development endeavours. Generally, the review is well written and insightful. The scientific community will benefit from such a review that covers a wide range of vaccine approaches and results. Below are my comments:

A-    Major comments:

1-    In the introduction, the sentence starts with “Clinical presentation…” lines 27,28 could be rephrased. The current sentence is quite unclear.

2-    First sentence in second paragraph in the introduction, line 39, please rephrase.

3-    Figure 1 is important, but could you please use colours or shaded boxes that show more contrast to differentiate between which trials were halted and which ones are active?

4-    Lines 107-113, would be better to mention why correlation between serum O-antigen IgG and protective efficacy in case of glycoconjugate and IgA in the case of oral live attenuated vaccine. Perhaps, mucosal vs systemic immune response?

5-    In 3.2.4 it was quite unclear what was the role of WRAIR if Pasteur Institute developed the vaccine. Was the role of WRAIR involved in testing in US adults?

6-    Paragraph lines 215-220, could you please explain more the “environmental enteric dysfunction”? it could be an important key to consider in developing/testing future vaccines in LMIC.

7-    Line 288, what are the six Shigella strains?

8-    Line 335, which vaccine was never commercialised, do you mean EcSf2a-2 and S. flexneri 2a O-Ag/rEPA?

9-    Line 356, it is unclear what promising results are the authors referring to, since ,from what it is understood, that Limmatech vaccine failed to meet primary endpoint of protection against all forms of shigellosis.

10- Any speculation why Limmatech vaccine did not work well since it is a glycoconjugate and glycoconjugate shown to be efficacious?

11- 4.1.4 what type of conjugate was used in order to develop bivalent glycoconjugate vaccine?

12- The authors refer to other reviews and papers, could they just summarise in a sentence or two the findings of the papers so to keep the reading flow?

Minor comments:

1-    A typo in line 353, compariosn (comparison)

Over all I thank the authors for such an insightful review.

Reviewer 2 Report

This paper is a review explaining the various pipelines of Shigella vaccines currently being studied. A general description of Shigella, Requirements for Shigella vaccines, and types of vaccines are well described. If there are only a few modifications, this is suitable for publication in the Special Issue.

1. In "Abstract", "Readouts of safety and immunogenicity from trials of lead candidate vaccines" are said to be available next year. If it is not confirmed, it would be better to express it in general terms than to indicate a specific time ('next year').

2. In "1. Introduction", it is better that the last sentence("This work complements articles by vaccine developers in this Vaccines Special Issue ‘Frontiers in Shigella Vaccine Development’. The reader is directed to these for specific details on the individual candidates.") be deleted. Or, in order for this thesis to become an independent thesis (without specifying that it is for a special issue), it is better to modify it using more objective or general words.

3. It seems that some of the contents of "2. Requirements for an effective Shigella vaccine" overlap with those of "1. Introduction" or "5. Discussion". Therefore, in "2. Requirements for an effective Shigella vaccine", it would be better to clearly organize by paragraph only the essential conditions that Shigella vaccine must have.
